# The Influence of NiTi Alloy on the Cyclic Fatigue Resistance of Endodontic Files

**DOI:** 10.3390/jcm9113755

**Published:** 2020-11-21

**Authors:** Celia Ruiz-Sánchez, Vicente Faus-Llácer, Ignacio Faus-Matoses, Álvaro Zubizarreta-Macho, Salvatore Sauro, Vicente Faus-Matoses

**Affiliations:** 1Department of Stomatology, Faculty of Medicine and Dentistry, University of Valencia, 46010 Valencia, Spain; celia.ruiz@uv.es (C.R.-S.); fausvj@uv.es (V.F.-L.); ignacio.faus@uv.es (I.F.-M.); vicente.faus@uv.es (V.F.-M.); 2Department of Endodontics, Faculty of Health Sciences, Alfonso X El Sabio University, 28691 Madrid, Spain; 3Department of Dentistry, Faculty of Health Sciences, CEU Cardenal Herrera University, 46115 Valencia, Spain; salvatore.sauro@uchceu.es; 4Department of Therapeutic Dentistry, I.M. Sechenov First Moscow State Medical University, 119146 Moscow, Russia

**Keywords:** endodontics, cyclic fatigue, NiTi alloy, M-Wire, CM-Gold Wire, CM-Blue Wire

## Abstract

Background: The aim of this study was to analyze the influence of NiTi alloy in endodontic rotary instruments on cyclic fatigue resistance. Methods: One hundred and sixty-four (164) sterile endodontic rotary files were selected and distributed into the following study groups: A: 25.08 F2 ProTaper Universal (PTU) (*n* = 41); B: 25.06 X2 ProTaper Next (PTN) (*n* = 41); C: 25.08 F2 ProTaper Gold (PTG) (*n* = 41), and D: 25.06 ProFile Vortex Blue (PVB) (*n* = 41). A cyclic fatigue device was designed to conduct the static cyclic fatigue tests with stainless steel artificial root canals systems with 250 µm apical diameter, 60° curvature angle, 5 mm radius of curvature, 20 mm length, and 6% (25.06) and 8% taper (25.08). Failure of the endodontic rotary instrument was detected by a single operator through direct observation and was also filmed to allow measurement of the exact time to failure. Results were analyzed using the ANOVA test and Weibull statistical analysis. Results: All pairwise comparisons presented statistically significant differences between the time to failure for the NiTi alloy study groups (*p* < 0.001), except between the PTN and PVB study groups (*p* = 0.379). In addition, statistically significant differences between the number of cycles to failure for the NiTi alloy study groups (*p* < 0.001) were also observed. Conclusions: The NiTi CM-Gold wire alloy of the ProTaper Gold endodontic rotary files resulted in greater resistance to cyclic fatigue than ProFile Vortex Blue, ProTaper Next, and ProTaper Universal endodontic rotary files.

## 1. Introduction

Endodontic rotary files have experienced continuous development since nickel–titanium (NiTi) files were introduced in the 1980s [1]. This alloy improved the flexibility and strength properties of endodontic rotary files compared with conventional stainless-steel endodontic instruments [2], and it simplified the treatment of root canals by improving the speed, accuracy, and safety of root canal shaping [3]. Despite continuous enhancements in the design and manufacture of NiTi endodontic rotary files to reduce the occurrence of complications during root canal treatment [4], failures remain a concern. Machado et al. retrospectively reported a fracture incidence of ProTaper Universal endodontic rotary files of 4.4% in 1031 teeth, mainly in mandibular first (8.8%) and second (9.6%) molars [5]; however, Bueno et al. reported no fractures in any of the 1104 Wave One Gold endodontic reciprocating files [6]. The more martensitic crystalline structure of the NiTi alloy of Wave One Gold endodontic reciprocating files allows higher flexibility and resistance than conventional austenitic endodontic rotary files. In addition, the removal of fractured fragments from NiTi endodontic rotary files within the root canal system is a challenge for the clinician because it prevents the disinfection of the entire root canal system [7].

Many variables can contribute to NiTi endodontic rotary file separation, such as root canal shape, instrument geometry, rotational speed, torque, sterilization cycles, the number of clinical uses, and the angle and radius of the curvature of the root canal system; however, researchers have focused their attention on the NiTi alloys, new surface treatments, and design improvements as prominent factors in the fracture resistance of NiTi endodontic rotary files [8,9,10,11]. Therefore, the surface treatments such as electropolishing, ion implantation, cryogenic treatment, and heat treatments improve the physical properties of NiTi endodontic rotary files, increasing their cyclic fatigue resistance [12]. Currently, the most widely used surface treatment is heat treatment, which consists of heat treating the NiTi alloy in a temperature range of around 450–550 °C and is carried out during or after the NiTi endodontic rotary file manufacturing process [12].

NiTi endodontic rotary systems are classified according to the crystal structure of the NiTi alloy: conventional NiTi, NiTi M-Wire, and R-Phase alloy endodontic rotary systems composed of an austenitic crystal structure, NiTi CM-Wire alloy endodontic rotary instruments composed of a martensitic crystal structure, and finally NiTi endodontic rotary systems that contain both austenitic and martensitic crystal structures [12]. Currently, novel endodontic rotary systems are developed with a higher concentration of the martensitic phase to improve the physical properties of hyperelasticity, shape memory, and fracture resistance compared to conventional NiTi alloys [12,13]. Conventional NiTi alloy of ProTaper Universal endodontic rotary files have lower cyclic fatigue resistance of endodontic rotary files compared to NiTi M-Wire of ProTaper Next endodontic rotary files [14,15,16,17,18]. Furthermore, NiTi CM-Wire alloy of Profile Vortex Blue and ProTaper Gold endodontic rotary files have shown a significantly (*p* < 0.05) higher cyclic fatigue resistance compared to the NiTi M-Wire alloy of endodontic rotary files [17,19,20,21,22,23].

The purpose of the present study was to analyze and compare the effect of the NiTi alloy on the static cyclic fatigue resistance of NiTi endodontic rotary files, with a null hypothesis (H_0_) stating that NiTi alloy of the endodontic rotary files would have no effect on the static cyclic fatigue resistance of NiTi endodontic rotary files.

## 2. Materials and Methods

### 2.1. Study Design

One hundred and sixty-four (164) sterile unused endodontic rotary files with a 250 µm apical diameter were used in this in vitro study. All NiTi endodontic rotary files were first inspected under magnification (OPMI pico, Zeiss, Oberkochen, Germany), and none were discarded. A controlled experimental trial was performed at the Department of Stomatology of the Faculty of Medicine and Dentistry at the University of Valencia (Valencia, Spain), between September 2019 and July 2020. The NiTi endodontic rotary files were categorized into the following study groups: A: 25.08 F2 conventional NiTi alloy ProTaper Universal (Dentsply Maillefer, Baillagues, Switzerland) (PTU) (*n* = 41); B: 25.06 X2 NiTi M-Wire alloy ProTaper Next (Dentsply Maillefer, Baillagues, Switzerland) (PTN) (*n* = 41); C: 25.08 F2 NiTi CM-Gold Wire alloy ProTaper Gold (Dentsply Maillefer, Baillagues, Switzerland) (PTG) (*n* = 41); and D: 25.06 NiTi CM-Blue Wire alloy ProFile Vortex Blue (Dentsply Tulsa Dental, Tulsa, OK, USA) (PVB) (*n* = 41).

### 2.2. Experimental Cycling Fatigue Procedure

Static fatigue procedures were performed through the cyclic fatigue device previously described [24,25,26] and based on the cyclic fatigue device developed by Plotino et al. [27]. The 16:1 reduction handpiece (X-Smart Plus, Dentsply Maillefer, Baillagues, Switzerland) was placed and fixed on a square polymethyl methacrylate structure using four fixations and the artificial root canals systems were placed in a square polymethyl methacrylate structure fixed by two fixations to the previous polymethyl methacrylate structure (Figure 1).

Two artificial root canal systems were constructed with 250 µm apical diameter, 60° curvature angle, 5 mm radius of curvature, 20 mm length, and 8% (25.08) (Figure 2A) and 6% taper (25.06) (Figure 2B). The artificial root canal systems were formed from a stainless-steel cylinder (Cocchiola S.A., Buenos Aires, Argentina) based on the anatomy of each NiTi endodontic rotary file to ensure an intimate contact between the artificial root canal system walls and the NiTi endodontic rotary files. The stainless-steel artificial root canal systems were partially removed to allow the identification of the exact time to failure of the tested endodontic rotary instruments (Figure 2A,B). Furthermore, all the static fatigue procedures were filmed to allow the measurement of the exact time to failure. The NiTi endodontic rotary files were placed inside the artificial root canal systems at their full working length before the static fatigue test started (Figure 2A,B).

The NiTi endodontic rotary files were used with a 16:1 reduction handpiece (X-Smart Plus, Dentsply Maillefer, Baillagues, Switzerland) according to the manufacturer’s instructions. The PTU, PTN and PTG endodontic rotary files were used at 300 revolutions per minute (rpm) and 5.2 N/cm torque; however, PVB endodontic rotary files were used at 500 rpm and 2.8 N/cm torque. The NiTi endodontic rotary files were used until fracture occurred in order to analyze the time to failure and the number of cycles to failure were measured and recorded with a digital chronometer (Timex, Middlebury, CT, USA).

To reduce the friction between the reciprocating files and the artificial canal walls, high-flow synthetic oil designed for the lubrication of the artificial root canal systems (Singer All-Purpose Oil; Singer Corp., Barcelona, Spain) was applied [28,29].

### 2.3. Scanning Electron Microscopy and Energy Dispersive X-ray Spectroscopy Analysis

A Scanning Electron Microscopy (SEM) and an Energy Dispersive X-Ray Spectroscopy (EDX) analysis were performed in the Department of Mechanical, Energetic, and Materials Engineering of the School of Industrial Engineering of the University of Extremadura (Badajoz, Spain) to perform a surface characterization and analyze the elemental composition of the chemical elements of the NiTi endodontic rotary files used in the static fatigue tests by means of the atomic weight percent measurement at three different locations (1, 2 and 3). SEM analysis (HITACHI S-4800, Fukuoka, Japan) of the NiTi endodontic rotary files was performed with the following exposure parameters: acceleration voltage: 20 kV, magnification from 100× to 6500× and a resolution between −1.0 nm at 15 kV and 2.0 nm at 1 kV.

### 2.4. Statistical Tests

Statistical analysis was performed by means of SAS 9.4 (SAS Institute Inc., Cary, NC, USA). Descriptive analysis included the mean and standard deviation (SD) for quantitative data. Comparative statistics was carried out by comparing the time to failure (minutes) and the number of cycles to failure using the ANOVA test. Weibull statistical analysis was also conducted. Descriptive analysis of the SEM and EDX analysis of the endodontic rotary files was also described. Statistical significance level was established at *p <* 0.05.

## 3. Results

The mean and SD values for time to failure (minutes) for each of the study groups are displayed in Table 1 and Figure 3.

The ANOVA analysis showed statistically significant differences between time to failure of PTG and PTN (*p* < 0.001), PTG and PTU (*p* < 0.001), PTG and PVB (*p* = 0.001), PTN and PTU (*p* < 0.001), and PTU and PVB (*p* < 0.001) NiTi endodontic rotary file study groups (Figure 3). However, no statistically significant differences were observed between time to failure of PTN and PVB (*p* = 0.379) (Figure 3).

The scale distribution parameter (η) of Weibull statistics found statistically significant differences between the time to failure of PTG and PVB (*p* = 0.003), PTU and PVB (*p* < 0.001), PTG and PTU (*p* < 0.001), PTN and PTU (*p* < 0.001), and PTG and PVB (*p* < 0.001) NiTi endodontic rotary files (Table 2, Figure 4); however, there were no statistically significant differences between the time to failure of PTN and PVB (*p* = 0.067) NiTi endodontic rotary file study groups (Table 2, Figure 4). The shape distribution parameter (β) of Weibull statistics found statistically significant differences between the time to failure of PTU and PVB (*p* < 0.001), PTG and PTU (*p* = 0.001), and PTN and PTU (*p* = 0.003) NiTi endodontic rotary files (Table 2, Figure 4). However, no statistically significant differences were observed between the time to failure of PTG and PVB (*p* = 0.467), PTN and PVB (*p* = 0.228), and PTG and PVB (*p* = 0.628) (Table 2, Figure 4) study groups. Briefly, the behavior of the endodontic rotary systems was very predictable, because most endodontic rotary files of each endodontic rotary system fractured at almost the same time. The less-steep slope generated by PTU NiTi endodontic rotary files indicates that the behavior is more predictable than in the other endodontic rotary systems, but they fracture earlier. The PTG NiTi endodontic rotary files showed a higher cyclic fatigue resistance than the PTU, PTN, and PVB NiTi endodontic rotary files.

The mean and SD values for number of cycles to failure for each of the study groups are displayed in Table 3 and Figure 5.

The ANOVA analysis showed statistically significant differences between the number of cycles to failure of PTG and PTN (*p* = 0.001), PTG and PTU (*p* < 0.001), PTG and PVB (*p* < 0.001), PTN and PTU (*p* < 0.001), PTN and PVB (*p* < 0.001), and PTU and PVB (*p* < 0.001) NiTi endodontic rotary file study groups (Figure 5).

The scale distribution parameter (η) of Weibull statistics found statistically significant differences between the number of cycles to failure of PTG and PVB (*p* < 0.001), PTN and PVB (*p* < 0.001), PTU and PVB (*p* < 0.001), PTG and PTU (*p* < 0.001), PTN and PTU (*p* < 0.001), and PTG and PVB (*p* < 0.001) NiTi endodontic rotary files (Table 4, Figure 6). The shape distribution parameter (β) of the Weibull statistics found statistically significant differences between the number of cycles to failure of PTU and PVB (*p* < 0.001), PTG and PTU (*p* < 0.001), and PTN and PTU (*p* = 0.003) NiTi endodontic rotary files (Table 4, Figure 6). However, no statistically significant differences were observed between the number of cycles to failure of PTG and PVB (*p* = 0.557), PTN and PVB (*p* = 0.228), and PTG and PVB (*p* = 0.535) (Table 4, Figure 6). Briefly, the behavior of the endodontic rotary systems is very predictable, because most endodontic rotary files of each endodontic rotary system break almost at the same time. The less steep slope generated by PTU NiTi endodontic rotary files indicates that the behavior is more predictable than in the other endodontic rotary systems, but they fracture earlier. The PVB NiTi endodontic rotary files showed a higher cyclic fatigue resistance than did the PTU, PTN, and PTG NiTi endodontic rotary files.

SEM analysis of the NiTi alloy PTU endodontic rotary files did not show accumulation of organic matter or structural alterations. Furthermore, manufacturing lines were distributed perpendicularly to the longitudinal axis of the endodontic rotary files and also parallel to each other due to the laser machining manufacturing process. The width and distance between the manufacturing lines corresponded to the precision and intensity of the laser machining process. Tubular porosity due to the laser machining process was also observed. Furthermore, all NiTi alloy PTU endodontic rotary files exhibited tubular porosity resulting from the combination of Ti alloys with other chemicals elements (Figure 7A–C).

EDX micro-analysis of PTU, PTN, PTG, and PVB NiTi endodontic rotary files was performed at 20 kV at three different locations, which allowed for a deep and accurate analysis of the NiTi endodontic rotary files composition. EDX micro-analysis at 20 kV showed that PTU NiTi endodontic rotary files were composed of C (2.17–3.18 wt.%), O (1.48–1.61 wt.%), Ti (42.72–43.25 wt.%), and Ni (52.56–52.97 wt.%) (Table 5 and Figure 8A–C).

SEM analysis of the NiTi M-Wire alloy PTN endodontic rotary files did not show accumulation of organic matter or structural alterations. Furthermore, manufacturing lines were distributed perpendicularly to the longitudinal axis of the endodontic rotary files and also parallel to each other due to the laser machining manufacturing process. Lower width and distance between the manufacturing lines was observed compared to the NiTi alloy PTU endodontic rotary files. Tubular porosity due to the laser machining process was also observed (Figure 9A–C).

EDX micro-analysis at 20 kV in three different locations showed that M-Wire alloy PTN endodontic rotary files were composed of C (3.31–4.43 wt.%), Al (0.56–1.39 wt.%), Ti (50.39–51.03 wt.%), and Ni (43.31–45.10 wt.%) (Table 6 and Figure 10A–C).

SEM analysis of the NiTi CM-Gold Wire alloy PTG endodontic rotary files did not show accumulation of organic matter or structural alterations. Furthermore, manufacturing lines were distributed perpendicularly to the longitudinal axis of the endodontic rotary files and also parallel to each other due to the laser machining manufacturing process (Figure 11A–C).

EDX micro-analysis at 20 kV in three different locations showed that CM-Gold Wire PTG endodontic rotary files were composed of C (2.11–2.20 wt.%), O (6.87–7.54 wt.%), Ti (40.49–40.82 wt.%), and Ni (49.59–50.20 wt.%) (Table 7 and Figure 12A–C).

SEM analysis of the CM-Blue Wire alloy PVB endodontic rotary files did not show accumulation of organic matter or structural alterations. Furthermore, manufacturing lines were distributed perpendicularly to the longitudinal axis of the endodontic rotary files and also parallel to each other due to the laser machining manufacturing process (Figure 13A–C).

EDX micro-analysis at 20 kV at three different locations showed that CM-Blue Wire PVB NiTi endodontic rotary files were composed of C (2.26–2.71 wt.%), O (11.37–13.30 wt.%), Ti (37.87–38.98 wt.%), and Ni (46.17–47.56 wt.%) (Table 8, Figure 14A–C).

## 4. Discussion

The results obtained in the present study reject the null hypothesis (H_0_) that states that the NiTi alloy of the endodontic rotary files has no effect on the static cyclic fatigue resistance of NiTi endodontic rotary files.

NiTi endodontic rotary files suffer unexpected fractures inside the root canal system despite their greater flexibility [4,30], which are produced by the stress caused by cyclic fatigue, torsional fatigue, or a combination of both [31,32]. The incidence of fracture of NiTi endodontic rotary files ranges from 0.09% to 5% [33,34] and influences the prognosis of the root canal system as the fractured fragment blocks the access to the apex, preventing disinfection of the root canal system. Furthermore, it has been reported that the presence of a previous periapical pathology combined with the fracture of an endodontic instrument represents a significant decrease in the success of the root canal treatment [35]. This is the reason why the cyclic fatigue resistance of NiTi endodontic rotary files has been widely analyzed.

In 2002, the American National Standard Institute and the American Dental Association established a standardization procedure to assess the cyclic and torsional fatigue resistance of stainless steel endodontic hand files [36]. The fatigue resistance was also described by the International Standards Organization (ISO) (ISO 3630/1) for 2% taper stainless steel endodontic hand files [37]; however, no specifications for the cyclic fatigue resistance of NiTi endodontic rotary systems above 2% taper exist, so many different static and dynamic cyclic fatigue devices have been developed to analyze the cyclic fatigue resistance of NiTi endodontic rotary files. A static cyclic fatigue device was used for this study because it allowed the analysis of an accumulative stress concentration located in the curvature area of the artificial root canal system that caused microstructural changes in the NiTi alloy responsible for the cyclic fatigue resistance of NiTi endodontic rotary files [38]. However, the static cyclic fatigue resistance comparison of different NiTi endodontic rotary systems is highly complicated due to difficulty in isolating its differences while respecting transversal designs, tapers, angular speeds, and NiTi alloy. For this reason, the most similar files for each endodontic rotary system were selected based on their length, apical diameter, cross section, angular movement, and speed and taper, although this represents a limitation within the study. In addition, the different speed values at which the NiTi endodontic rotary systems were used could influence the study results, but the NiTi endodontic rotary systems were used at the speed values recommended by the manufacturers. PTU, PTN, and PTG NiTi endodontic rotary systems were used at 300 rpm, while the PVB NiTi endodontic rotary system was used at 500 rpm; however, most authors agree that rotation speed appears to have no effect on the resistance of NiTi endodontic rotary systems [32,39,40].Gao et al. showed no statistical significant differences (*p* ˃ 0.05) between the NiTi endodontic rotary files made of the same NiTi alloy and the same apical diameter but operated at different rotation speeds to cyclic fatigue resistance [41].

Previous studies have also analyzed the influence of the NiTi alloy on the cyclic fatigue resistance of endodontic rotary files, and most agree that the martensitic phase of NiTi alloy is the crystalline structure most resistant to cyclic fatigue [42]. In this study, the comparison of the static cyclic fatigue resistance of conventional NiTi alloy of PTU endodontic rotary files (1.24 ± 0.21 min) and NiTi M-Wire alloy of PTN endodontic rotary files (2.63 ± 0.58 min) showed a significantly (*p* < 0.001) higher static cyclic fatigue resistance of PTN NiTi endodontic rotary files compared to PTU endodontic rotary files. In addition, the comparison between conventional NiTi alloy of PTU endodontic rotary files (1.24 ± 0.21 min) and NiTi CM-Gold Wire alloy of PTG endodontic rotary files (3.42 ± 0.85 min) was influenced by fewer variables compared to the rest of the NiTi endodontic rotary systems because they presented similar cross-sections and designs. These NiTi endodontic rotary systems showed a significantly (*p* < 0.001) higher static cyclic fatigue resistance for PTG NiTi endodontic rotary files than PTU endodontic rotary files, which highlights the influence of the NiTi alloy on the cyclic fatigue resistance of NiTi endodontic rotary files, as was already shown in previous studies [16,19]. Furthermore, the comparison between the NiTi CM-Gold Wire alloy of PTG endodontic rotary files (3.42 ± 0.85 min) and the NiTi CM-Blue Wire alloy of PVB endodontic rotary files (2.86 ± 0.82 min) also showed significantly (*p* < 0.001) higher static cyclic fatigue resistance of PTG NiTi endodontic rotary files than PVB endodontic rotary files; however, it is difficult to isolate the most relevant variable or the most determinant variable combination in cyclic fatigue resistance of NiTi endodontic rotary files. These results differ when comparing the cyclic fatigue resistance of the NiTi CM-Blue Wire alloy of Reciproc Blue endodontic reciprocating system with the NiTi CM-Gold Wire alloy from the Wave One Gold endodontic reciprocating system. The cyclic fatigue resistance of the Reciproc Blue endodontic reciprocating system showed a higher resistance to cyclic fatigue than the Wave One Gold endodontic reciprocating system, probably due to the cross-sectional design [43,44]. The influence of the cross-sectional design on the cyclic fatigue resistance was also analyzed by comparing the NiTi M-Wire alloy of the Reciproc endodontic reciprocating system with the NiTi M-Wire alloy of the Wave One endodontic reciprocating system and showed that the Reciproc endodontic reciprocating system had significantly (*p* < 0.001) higher static cyclic fatigue resistance than the Wave One endodontic reciprocating system [45].

The novel heat treatments of NiTi alloys show more resistance to cyclic fatigue than conventional NiTi alloys. Elanghy et al. reported a statistically significant (*p* < 0.001) higher cyclic fatigue resistance of ProTaper Next and ProTaper Gold compared to conventional NiTi alloys of TRUShape and ProTaper Universal NiTi endodontic rotary systems [46], and Uygun et al. reported similar results when comparing ProFile Vortex Blue and ProTaper Next with the conventional NiTi alloy of ProTaper Universal [47]. These results agreed with those obtained in the present study and highlight the relevance of the alloying elements in the allotropic transformation of the crystalline structure of the NiTi alloy, which in turn influences the physical and mechanical properties of the endodontic rotary files [47]. Titanium is an allotropic metal which can present two crystalline structures: compact hexagonal (α or austenite) and body-centered cubic (β or martensite). Depending on the stabilizing effect of the α and β phases, the alloying elements of titanium are classified as neutral elements, betagenic elements, or stabilizers of the α phase and alphagenic elements or stabilizers of the β phase. The stabilization of the phases implies the increase or decrease of the transition temperature β [48]; specifically, alphagenic elements increase the transition temperature β. Among the alphagenic elements, Al is the most important alloying element, although O, C, and N can also be used. The results obtained from the EDX micro-analysis showed the presence of the above mentioned alloying elements. The atomic weight of the alloying elements showed an upward trend from the conventional NiTi alloy of PTU endodontic rotary files (C (2.17–3.18 wt.%) and O (1.48–1.61 wt.%)), to the NiTi M-Wire alloy of PTN endodontic rotary files (C (3.31–4.43 wt.%) and Al (0.56–1.39 wt.%)), to the NiTi CM-Blue Wire alloy of PVB endodontic rotary files (C (2.26–2.71 wt.%) and O (11.37–13.30 wt.%)), and finally to the NiTi CM-Gold Wire alloy of PTG endodontic rotary files (C (2.11–2.20 wt.%) and O (6.87–7.54 wt.%)); this leads to more crystalline structures that are more martensitic and therefore more flexible and resistant to fracture, which is consistent with the time to failure observed in the study of resistance to static cyclic fatigue.

## 5. Conclusions

NiTi CM-Gold wire alloy of the ProTaper Gold endodontic rotary files is more resistance to cyclic fatigue than ProFile Vortex Blue, ProTaper Next, and ProTaper Universal endodontic rotary files, due to the elements of the alloy present in the crystalline structure that give the ProTaper Gold endodontic rotary files greater flexibility and resistance.

## Figures and Tables

**Figure 1 jcm-09-03755-f001:**
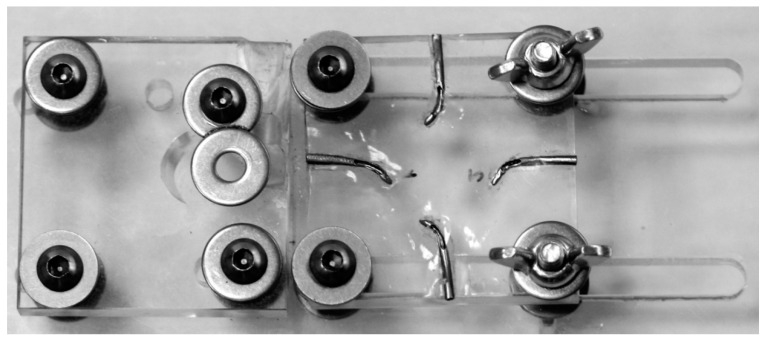
Static cyclic fatigue device with embedded stainless steel artificial root canal systems.

**Figure 2 jcm-09-03755-f002:**
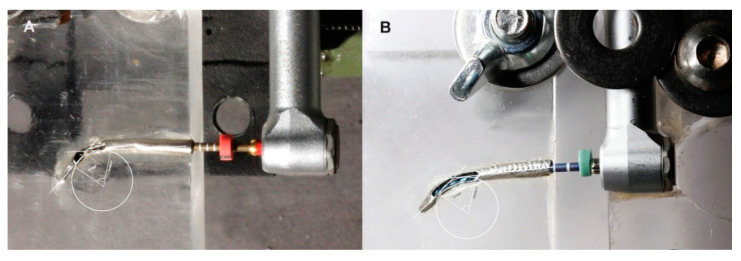
(**A**) ProTaper Gold (PTG) endodontic rotary file inside the 8% taper artificial root canal system and (**B**) ProFile Vortex Blue (PVB) endodontic rotary file inside the 6% taper artificial root canal system.

**Figure 3 jcm-09-03755-f003:**
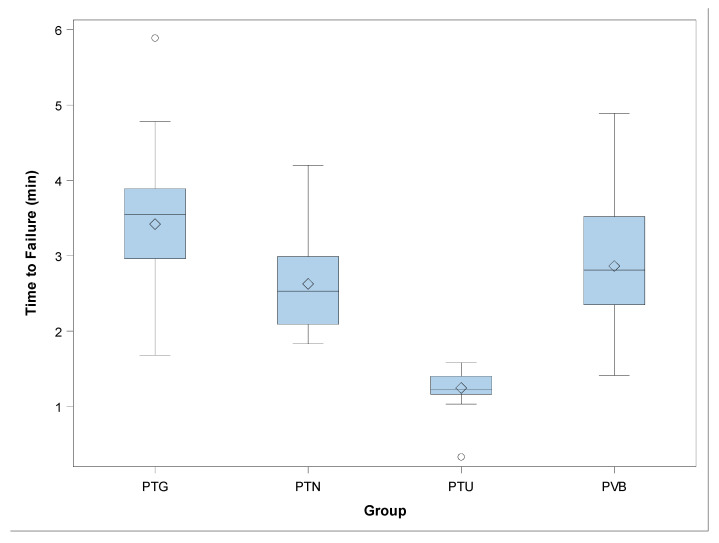
Box plot of the time to failure. The horizontal line in each box represents the respective median value of the study groups. ◊; Mean value of the box plots. PTU: ProTaper Universal; PTN: ProTaper Next; PTG: ProTaper Gold; PVB: ProFile Vortex Blue.

**Figure 4 jcm-09-03755-f004:**
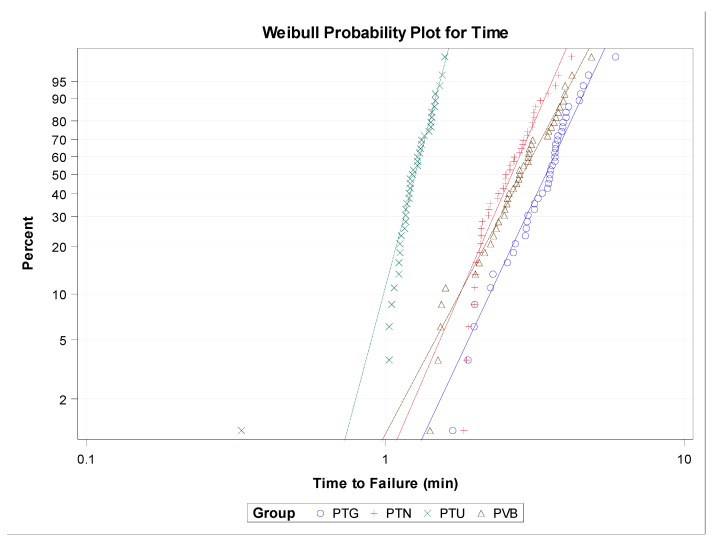
Weibull probability plot of time to failure for the study groups.

**Figure 5 jcm-09-03755-f005:**
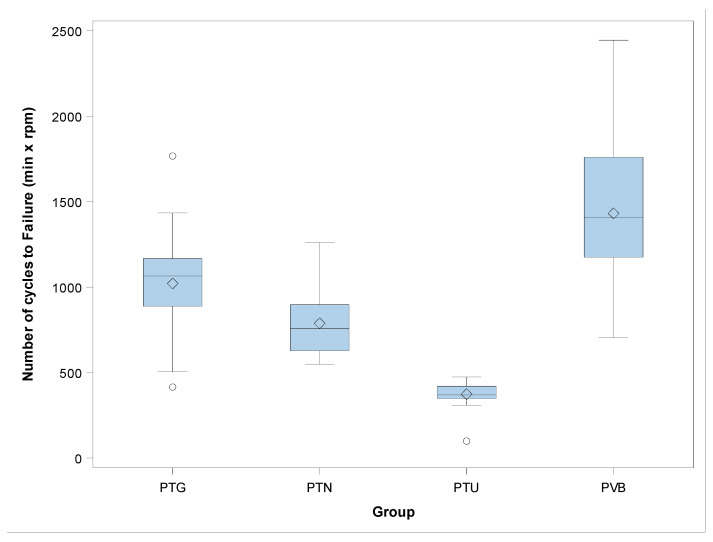
Box plot of the number of cycles to failure. The horizontal line in each box represents the respective median value of the study groups.○: Mean value of the box plots.

**Figure 6 jcm-09-03755-f006:**
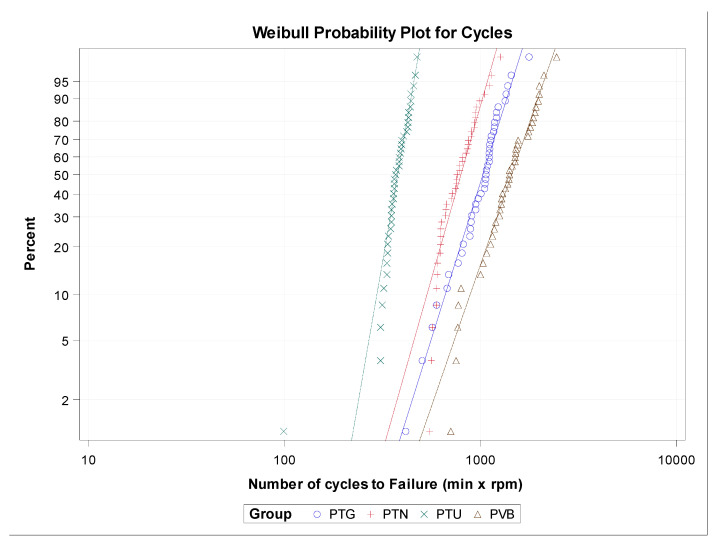
Weibull probability plot of the number of cycles to failure for the study groups.

**Figure 7 jcm-09-03755-f007:**
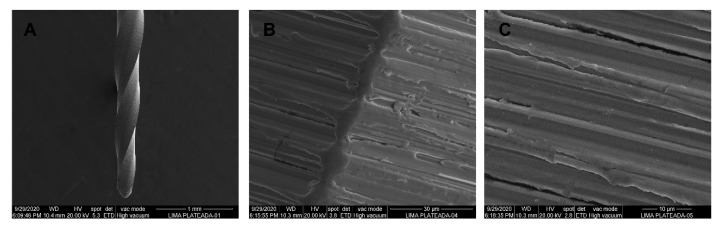
SEM images of NiTi alloy ProTaper Universal (PTU) endodontic rotary files at (**A**) 100×, (**B**) 3600×, and (**C**) 6500×.

**Figure 8 jcm-09-03755-f008:**
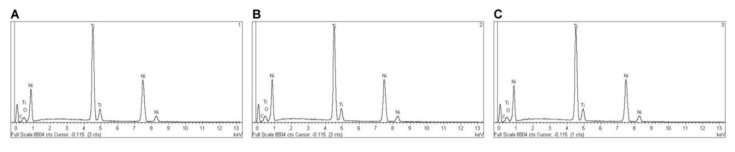
EDX micro-analysis of NiTi alloy PTU endodontic rotary files in location (**A**) 1, (**B**) 2, and (**C**) 3.

**Figure 9 jcm-09-03755-f009:**

SEM images of M-Wire alloy ProTaper Next (PTN) endodontic rotary files at (**A**) 100×, (**B**) 3600×, and (**C**) 6500×.

**Figure 10 jcm-09-03755-f010:**
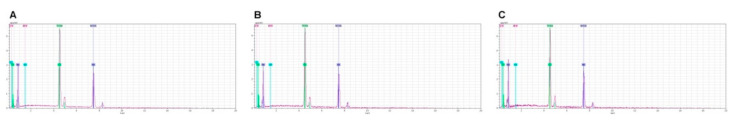
EDX micro-analysis of M-Wire alloy PTN endodontic rotary files at locations (**A**) 1, (**B**) 2, and (**C**) 3.

**Figure 11 jcm-09-03755-f011:**
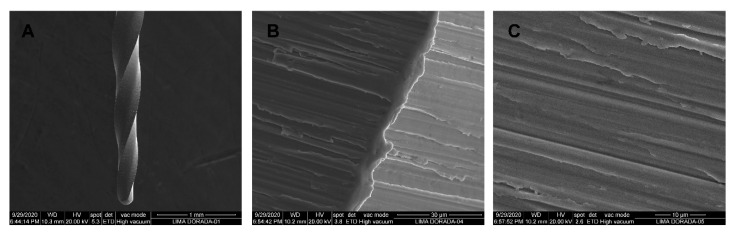
SEM images of CM-Gold Wire alloy PTG endodontic rotary files at (**A**) 100×, (**B**) 3600×, and (**C**) 6500×.

**Figure 12 jcm-09-03755-f012:**
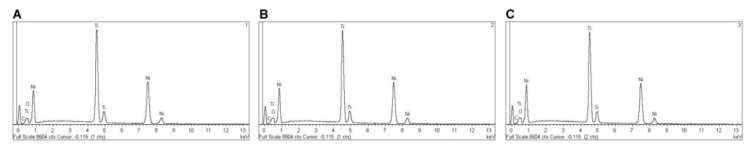
EDX micro-analysis of CM-Gold Wire PTG endodontic rotary files in location (**A**) 1, (**B**) 2, and (**C**) 3.

**Figure 13 jcm-09-03755-f013:**
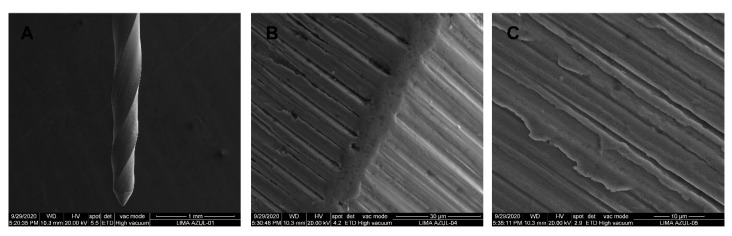
SEM images of CM-Blue Wire PVB endodontic rotary files at (**A**) 100×, (**B**) 3600×, and (**C**) 6500×.

**Figure 14 jcm-09-03755-f014:**
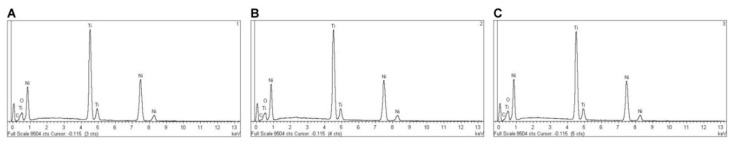
EDX micro-analysis of CM-Blue Wire PVB endodontic rotary files at locations (**A**) 1, (**B**) 2, and (**C**) 3.

**Table 1 jcm-09-03755-t001:** Descriptive analysis of the time to failure (minutes).

Study Group	*n*	Mean	SD	Minimum	Maximum
PTU	41	1.24 ^a^	0.21	0.33	1.58
PTN	41	2.63 ^b^	0.58	1.83	4.20
PTG	41	3.42 ^c^	0.85	1.68	5.89
PVB	41	2.86 ^b^	0.82	1.41	4.89

^a,b,c^ Statistically significant differences between groups (*p* < 0.05). PTU: ProTaper Universal; PTN: ProTaper Next; PTG: ProTaper Gold; PVB: ProFile Vortex Blue.

**Table 2 jcm-09-03755-t002:** Weibull statistics of time to failure of the study groups.

Study Group	Weibull Shape (β)				Weibull Scale (η)			
	Estimate	St Error	Lower	Upper	Estimate	St Error	Lower	Upper
PTU	7.7054	0.9453	6.0585	9.8000	1.3203	0.0279	1.2666	1.3762
PTN	4.7196	0.5369	3.7764	5.8983	2.8604	0.1005	2.6701	3.0643
PTG	4.3641	0.5001	3.4862	5.4632	3.7429	0.1414	3.4758	4.0307
PVB	3.8689	0.4629	3.0601	4.8914	3.1652	0.1350	2.9114	3.4410

**Table 3 jcm-09-03755-t003:** Descriptive statistics of the number of cycles to failure of the study groups.

Study Group	*n*	Mean	SD	Minimum	Maximum
PTU	41	373.46 ^a^	62.40	99.00	474.00
PTN	41	787.90 ^b^	173.91	549.00	1260.00
PTG	41	1021.49 ^c^	264.81	415.00	1767.00
PVB	41	1431.46 ^d^	411.60	705.00	2445.00

^a,b,c,d^ Statistically significant differences between groups (*p* < 0.05).

**Table 4 jcm-09-03755-t004:** Weibull statistics of the number of cycles to failure for the study groups.

Study Group	Weibull Shape (β)				Weibull Scale (η)			
	Estimate	St Error	Lower	Upper	Estimate	St Error	Lower	Upper
PTU	7.7054	0.9453	6.0585	9.8000	396.0883	8.3847	379.9908	412.8678
PTN	4.7196	0.5369	3.7764	5.8983	858.1237	30.1478	801.0235	919.2942
PTG	4.2673	0.4950	3.3995	5.3566	1119.912	43.1923	1038.377	1207.8487
PVB	3.8689	0.4629	3.0601	4.8914	1582.588	67.4766	1455.711	1720.5229

**Table 5 jcm-09-03755-t005:** Mean atomic weight percent (%) of NiTi alloy PTU endodontic rotary files at 15 kV and 20 kV and three different locations (1, 2 and 3).

Spectrum	C	O	Ti	Ni
PTU 20 kV (1)	2.17	1.61	43.25	52.97
PTU 20 kV (2)	3.18	1.54	42.72	52.56
PTU 20 kV (3)	2.65	1.48	42.99	52.87
Sigma	0.55	0.65	0.46	0.53

**Table 6 jcm-09-03755-t006:** Mean atomic weight percent (%) of M-Wire alloy PTN endodontic rotary files at 15 kV and 20 kV and three different locations (1, 2, and 3).

Spectrum	C	Al	Ti	Ni
PTN 20 kV (1)	4.43	1.24	51.03	43.31
PTN 20 kV (2)	3.31	1.39	50.39	44.90
PTN 20 kV (3)	3.43	0.56	50.91	45.10
Sigma	0.61	0.44	0.34	0.98

**Table 7 jcm-09-03755-t007:** Mean atomic weight percent (%) of CM-Gold Wire alloy PTG endodontic rotary files at 15 kV and 20 kV at three different locations (1, 2 and 3).

Spectrum	C	O	Ti	Ni
PTG 20 kV (1)	2.11	6.87	40.82	50.20
PTG 20 kV (2)	2.12	7.35	40.49	50.04
PTG 20 kV (3)	2.20	7.54	40.67	49.59
Sigma	0.54	0.67	0.44	0.53

**Table 8 jcm-09-03755-t008:** Mean atomic weight percent (%) of CM-Blue Wire PVB endodontic rotary files at 15 kV and 20 kV at three different locations (1, 2 and 3).

Spectrum	C	O	Ti	Ni
PVB 20 kV (1)	2.66	13.30	37.87	46.17
PVB 20 kV (2)	2.26	11.80	38.38	47.56
PVB 20 kV (3)	2.71	11.37	38.98	46.94
Sigma	0.58	0.69	0.44	0.53

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
