# Peer review of "The Influence of NiTi Alloy on the Cyclic Fatigue Resistance of Endodontic Files"

_jcm, 2020, doi:10.3390/jcm9113755_

Round 1
Reviewer 1 Report
This study is interesting for analyzing cyclic fatigue resistance of various endodontic rotary files. If you revise some minor issues, then it will be appropriate for publishing in JCM.
- Please clarify the XXXX in line 64-67.
- In table 1, the superscript in PVB group can be changed to "b", because no statistical differences were observed between time to failure of PTN and PVB.
- In table 2 and table 4, the results of Weibull statistics are difficult to understand. Please clarify the results and reduce them shortly.
The SEM and EDX analyses were well described.
Author Response
Dear Reviewer 1,
I’m pleased to resubmit the manuscript of the work entitled, “Novel Cyclic Fatigue Device to Analyze the Cyclic Fatigue Resistance of Conventional NiTi Alloy, NiTi M-Wire Alloy, NiTi CM-Gold Wire Alloy and NiTi CM-Blue Wire Alloy Endodontic Rotary Files”
Reviewer 1: English language and style are fine/minor spell check required
Response: In order to adapt to the reviewer's 1 comments, we have send the manuscript to the English Editing Service of MDPI. We attached the Certificate.
Reviewer 1: Please clarify the XXXX in line 64-67.
Response: In order to adapt to the reviewer's 1 comments, we have clarified the incorrect sentence.
Reviewer 1: In table 1, the superscript in PVB group can be changed to "b", because no statistical differences were observed between time to failure of PTN and PVB.
Response: In order to adapt to the reviewer's 1 comments, we have corrected the superscript.
Reviewer 1: In table 2 and table 4, the results of Weibull statistics are difficult to understand. Please clarify the results and reduce them shortly.
Response: In order to adapt to the reviewer's 1 comments, we have clarify the Weibull statistics.
We take this opportunity to thank the recommendations and suggestions made by the reviewers to improve the document.
Yours sincerely
Reviewer 2 Report
Conclusions need to be improved by the authors with the data provided in the results part
Author Response
Dear Reviewer 2,
I’m pleased to resubmit the manuscript of the work entitled, “Novel Cyclic Fatigue Device to Analyze the Cyclic Fatigue Resistance of Conventional NiTi Alloy, NiTi M-Wire Alloy, NiTi CM-Gold Wire Alloy and NiTi CM-Blue Wire Alloy Endodontic Rotary Files”
Reviewer 2: English language and style are fine/minor spell check required
Response: In order to adapt to the reviewer's 2 comments, we have send the manuscript to the English Editing Service of MDPI. We attached the Certificate.
Reviewer 2: Conclusions need to be improved by the authors with the data provided in the results part
Response: In order to adapt to the reviewer's 2 comments, we improved the Conclusion section.
We take this opportunity to thank the recommendations and suggestions made by the reviewers to improve the document.
Yours sincerely
Reviewer 3 Report
This is an overall well done paper looking at the cyclic fatigue resistance of a few types of files. There is nothing really that novel about this study but it is technically well done and well written and sound conclusions are drawn. I have a few suggestions to the authors that I anticipate can be fixed and lead to publication
The title is rather long. This is ultimately up to the authors to decide, I respect that, but I would suggest shortening it.
Some statistics in the introduction, and not later, about frequency/rate of file fracture would be highly useful.
A clearer explanation of what a martensitic phase in the introduction is would be good for more general readers.
It is not clear how the device fabricated is novel as the authors cite 4 papers that they based the design on. If this is not clarified I suggest removing that portion from the title.
The second to last paragraph of the introduction has missing information.
Fig 2- the angle measurement is hard to read.
Use of synthetic oil – The authors should note the commonness of using synthetic oil as a lubricant in these sorts of tests.
SEMs – I don’t find these add very much to the story. The authors repeatedly state “ manufacturing lines are distributed perpendicularly to the longitudinal axis of the endodontic rotary files and also parallel to each other due to the manufacturing process by laser machining.” Can the authors draw more conclusions from the SEMs?
Conclusion – “The conclusion derived from the present study is” – Delete this.
Author contributions looks unfinished.
Statistics and analysis are standard and fine.
Author Response
Dear Reviewer 3,
I’m pleased to resubmit the manuscript of the work entitled, “Novel Cyclic Fatigue Device to Analyze the Cyclic Fatigue Resistance of Conventional NiTi Alloy, NiTi M-Wire Alloy, NiTi CM-Gold Wire Alloy and NiTi CM-Blue Wire Alloy Endodontic Rotary Files”
Reviewer 3: English language and style are fine/minor spell check required
Response: In order to adapt to the reviewer's 3 comments, we have send the manuscript to the English Editing Service of MDPI. We attached the Certificate.
Reviewer 3: The title is rather long. This is ultimately up to the authors to decide, I respect that, but I would suggest shortening it.
Reviewer 3: Some statistics in the introduction, and not later, about frequency/rate of file fracture would be highly useful.
Response: In order to adapt to the reviewer's 3 comments, we have added the fracture incidence of endodontic rotary file.
Reviewer 3: A clearer explanation of what a martensitic phase in the introduction is would be good for more general readers.
Response: In order to adapt to the reviewer's 3 comments, we have added an explanation of the influence of the martensitic crystalline structure on the resistance of endodontic files according Reviewer 3 suggestions
Reviewer 3: It is not clear how the device fabricated is novel as the authors cite 4 papers that they based the design on. If this is not clarified I suggest removing that portion from the title.
Response: In order to adapt to the reviewer's 3 comments, we have changed the title.
Reviewer 3: The second to last paragraph of the introduction has missing information
Response: In order to adapt to the reviewer's 3 comments, we have we have clarified the incorrect sentence.
Reviewer 3: Fig 2- the angle measurement is hard to read.
Response: In order to respond to the reviewer's 3 comments, we would like to apologize because we do not have a better image.
Reviewer 3: Use of synthetic oil – The authors should note the commonness of using synthetic oil as a lubricant in these sorts of tests
Response: In order to adapt to the reviewer's 3 comments, we added 2 references were synthetic oil was used.
Reviewer 3: SEMs – I don’t find these add very much to the story. The authors repeatedly state “ manufacturing lines are distributed perpendicularly to the longitudinal axis of the endodontic rotary files and also parallel to each other due to the manufacturing process by laser machining.” Can the authors draw more conclusions from the SEMs?
Response: In order to respond to the reviewer's 3 comments, we clarify that the manufacturing process, as well as surface defects resulting from the manufacturing process, can influence the fracture resistance of endodontic rotary files [Keskin NB, Inan U. Cyclic fatigue resistance of rotary NiTi instruments produced with four different manufacturing methods. Microsc Res Tech. 2019 Oct;82(10):1642-1648.]. For this reason, the authors carried out the SEM analysis to analyze the manufacturing process of the files and discard that they had manufacturing defects imperceptible to the naked eye.
Reviewer 3: Conclusion – “The conclusion derived from the present study is” – Delete this.
Response: In order to adapt to the reviewer's 3 comments, we removed this sentence.
Reviewer 3: Author contributions looks unfinished.
Response: In order to adapt to the reviewer's 3 comments, we have completed the Author Contributions.
We take this opportunity to thank the recommendations and suggestions made by the reviewers to improve the document.
Yours sincerely,